# Structural Features of Monoethanolamine Aqueous Solutions with Various Compositions: A Combined Experimental and Theoretical Study Using Vibrational Spectroscopy

**DOI:** 10.3390/molecules28010403

**Published:** 2023-01-03

**Authors:** Sergey A. Legkov, Galina N. Bondarenko, Julia V. Kostina, Eduard G. Novitsky, Stepan D. Bazhenov, Alexey V. Volkov, Vladimir V. Volkov

**Affiliations:** A. V. Topchiev Institute of Petrochemical Synthesis Russian Academy of Sciences, 119991 Moscow, Russia

**Keywords:** monoethanolamine, absorption, FTIR-spectroscopy, quantum chemical calculations, associates

## Abstract

In this work, we studied aqueous solutions of monoethanolamine (MEA), which are widely used to remove CO_2_ from flue and oil gases. This study combined experimental and theoretical methods of vibrational spectroscopy, using high-temperature infrared spectroscopy, quantum-chemical calculations of theoretical vibrational spectra, and structural electronic and energy characteristics of model structures. MEA has a propensity to form associations between various compositions and structures with water molecules, as well as those composed solely of water molecules. The structural and energy characteristics of such associates were analyzed in terms of their ability to interact and retain carbon dioxide. The influence of elevated temperatures and concentration of aqueous MEA solution on change in the structure of associates has also been investigated. An analysis of theoretical and experimental vibrational spectra allowed us to examine the IR spectra of MEA solutions, and identify the bands responsible for the formation of associates that would sorb CO_2_ well, but would delay its desorption from the solution.

## 1. Introduction

The absorption method of acidic components (CO_2_, SO_2_, CS_2_, etc.) removal from processed gases using aqueous solutions of ethanolamines, which has been used for almost one hundred years [1] in oil and gas industry at low impurity concentrations, still lacks alternative solutions [2,3]. The technology involved in carbon dioxide absorption from flue, oil or associated gases via aqueous solutions of ethanolamines, primarily monoethanolamine (MEA), consists of two successive processes: absorption of CO_2_ from a gas mixture and subsequent desorption of CO_2_ at the regeneration stage of the absorbent. The first stage is carried out at moderate temperatures (30–50 °C), and the second at rather high temperatures (120–140 °C). This determines the main drawback of this technology, since at elevated temperatures, intense thermal and thermo-oxidative degradation of MEA occurs [4,5,6,7]. Decomposition of MEA leads to significant economic losses due to a decrease in its absorption capacity and accelerated corrosion of equipment.

The lack of a widely used gas removal technology has inspired many studies on the analysis of MEA decomposition products. Dozens of different chemical compounds, found in MEA decomposition products, using modern physicochemical research methods [8,9,10,11,12], do not clarify the mechanism of transformation of MEA in aqueous solutions, in the presence of carbon dioxide. The authors of [13] concluded that IR Fourier spectroscopy is the most informative method for evaluating the products of the transformation of alkanolamines in aqueous solutions saturated with carbon dioxide. In [14,15], based on the study of specific electrical conductivity of alkanolamine solutions, it was suggested that both initial solutions and those saturated with CO_2_ aqueous MEA have a complex structure, forming various associations. Association that occurs in MEA solutions, according to these authors, determines the reason for the decrease in absorption capacity of MEA, and the need to regenerate the absorbent during operation.

In this work, we studied aqueous solutions of MEA at various concentrations at elevated temperatures. The aim of this study was to develop a method based on a comparison of experimental IR spectra (including high-temperature spectra) and theoretical vibrational spectra of corresponding models obtained from quantum-chemical calculations. This approach enables the reliable study of the structure of associates, formed during the interaction, and allows us to identify the supramolecular structures of these solutions (associates, micelles, etc.), as well as to study the mechanism of carbon dioxide sorption and desorption processes.

## 2. Results and Discussion

### Structure and Vibrational Spectra of Aqueous Solutions of MEA

Since aqueous solutions of MEA are used in technological processes to remove CO_2_ from processed gases, it is necessary to establish the effect of the solvent (water) and absorbate (CO_2_) on the structural characteristics of MEA. The most promising research method for solving this problem is a combination of experimental and theoretical methods of vibrational spectroscopy, which make it possible to interpret recorded IR spectra of initial aqueous MEA solutions and those saturated with CO_2_.

Figure 1a shows the IR spectrum of initial (undissolved) MEA, and Figure 1b shows a series of IR spectra of aqueous MEA solutions with different concentrations. There is practically no similarity between these spectra, even considering that in the spectra of Figure 1b, a very intense broad band with a maximum of 3408 cm^−1^ and a band of medium intensity at 1655 cm^−1^, are associated, respectively, with valence and deformation vibrations of water. At the same time, the spectra of aqueous solutions (Figure 1b) show significant changes in intensities and certain shifts of the bands in spectra of different concentrations.

Quantum-chemical calculations of an isolated MEA molecule in vacuum and water indicate the presence of two stable isomers (Figure 2a,b), one of which (a) has an almost flat transoidal structure (dihedral angle O-C-C-N is 178°), while the second (b), which is only 0.48 kcal/mol more favorable in terms of energy, has a non-planar gauche structure with a dihedral angle Q (O-C-C-N) = 64°.

The theoretical spectrum of an individual molecule, shown in Figure 3, also bears little resemblance to the experimental spectrum in Figure 1. However, theoretical frequencies of 1060, 1395, and 1450 cm^−1^ coincide quite well in terms of the maximum with experimental MEA bands (Figure 1a). An analysis of vibrational modes in the theoretical MEA spectrum, which allows us to interpret IR spectrum bands, shows that frequency 1060 cm^−1^ is associated with stretching C-O bond (νC-O), and the other two frequencies (1395 and 1450 cm^−1^) refer to the bending of two CH_2_ groups (δCH_2_) in the MEA structure. All other theoretical frequencies—especially frequencies of the stretching and bending vibrations of the O-H and N-H bonds—differ significantly from the bands in experimental spectrum in Figure 1 both in terms of the positions of the maxima and in their intensity. Such differences in the theoretical and experimental IR spectra of MEA indicate that MEA molecules cannot exist as individual molecules. Instead, they exist in the form of associates in a liquid phase and in aqueous solutions.

Calculations show the possibility of MEA associate existence as dimers and linear trimers, with the formation of O-H …N hydrogen bonds between individual MEA molecules. The lengths of such hydrogen bonds are 2.1 Å in dimer and 2.3 Å in trimer forms. In this case, the association energy drops from 3.73 kcal/mol in the dimer to 3.28 kcal/mol in the trimer. According to the calculations, MEA tetramer already has a more complex structure: three molecules form a cyclic associate, and the fourth molecule is not associated with a cyclic trimer. Theoretical vibrational spectra of such associates are already in good agreement with the experimental IR spectrum of MEA shown in Figure 1a, especially if we compare the experimental IR spectrum with the superposition of theoretical spectra of two or three different models of associates.

MEA’s tendency to associate with water molecules is highly pronounced. Table 1 presents hydrogen bond lengths and association energies in these complexes, illustrating features of the presented structures of the complexes. Moreover, as calculations show, the association energy of MEA with water molecules is higher than the association energy of MEA⋯MEA (see Table 2). In theoretical spectra of such associates, there is a noticeable decrease in intensities of bands in the region 400–1600 cm^−1^, i.e., in the area stretching vibration of MEA and water, and a strong increase in the intensity of bands in the region of 3200–3600 cm^−1^ from stretching O-H and N-H bonds, which is in good agreement with the experimental spectra of aqueous MEA solutions (Figure 1b). Figure 4 shows the structures of MEA that associate with two water molecules (a) and associations between three MEA molecules with three water molecules. A structural feature of such associate complexes is the tendency to form hydrogen bonds between MEA molecules and water molecules, as well as the association between water molecules. In this case, association between MEA molecules in complex “b” does not occur, although this complex contains three MEA molecules. In aqueous MEA solutions, there is a tendency for individual MEA molecules to be surrounded by water molecules. 

Table 2 presents structural (bond lengths, valence angles H-N-H, and dipole moments), electronic (charges on N, O, H atoms), and energy (energy of association in associate complexes) characteristics, as well as the lengths of hydrogen bonds in associates (for complex associates, the shortest hydrogen bonds are presented). In the first two lines of the table, the characteristics for the initial molecules (MEA and H_2_O) are presented for comparison.

High stability of the MEA-10H_2_O associate complex (Figure 5) is due to a whole system of hydrogen bonds that arise between the oxygen and hydrogen atoms of water molecules, with lengths ranging from 1.9 to 1.7 Å.

The MEA molecule is also included in the hydrogen bonding system, with the shortest hydrogen bond, 1.693 Å, occurring between the O1 oxygen atom of MEA and the H41 hydrogen atom of the nearest water molecule. The MEA nitrogen atom is also linked to the H14 hydrogen atom from a water molecule via a 1.891 Å hydrogen bond. The entire associate complex is a three-way cycle, in which two cycles are composed exclusively of alternating valence and hydrogen O-H bonds of water molecules, and the distances between two oxygen atoms not bound by hydrogen bonds in these cycles are in a range 4.2–4.5 Å. The third cycle includes five associated water molecules and a MEA molecule. In this cycle, rather short distances (2.7 Å) between oxygen O1…O18 and O1…O39 atoms are realized. Notably, between H11 and O33 atoms, there is an internal hydrogen bond (2.5 Å). Despite the complexity of its structure, this associate has fairly high symmetry and its dipole moment has a small value of 2.825D. The nitrogen atom in the MEA molecule in the MEA-10H_2_O associate complex has a well-defined tetrahedral environment, since the H14-N4-C2 and H10-N4-H11 atoms lie in mutually perpendicular planes. Notably, in the initial MEA molecule, in all associates presented in Table 2 and other calculated models of associates (MEA-4H_2_O, MEA-8H_2_O), there is a trigonal pyramid structure around the nitrogen atom, at the top of which lies a nitrogen atom. At the base lies a carbon atom and two hydrogen atoms.

An associate of 10 water molecules (Figure 6) also has highly polarized O-H covalent bonds (Table 2) and 12 hydrogen bonds, with lengths ranging from 1.6 to 2.2 Å. As an associate complex of MEA-10H_2_O, this associate is also a three-way cycle. Two of the cycles include five and six water molecules, and the last one includes four water molecules. The diameter values of these cycles range from 4.8 to 5.5 Å. Considering that a linear carbon dioxide molecule has a length of 2.3 Å, one can imagine that CO_2_ molecules will easily enter the network formed by associates of 10 water molecules and can also easily leave it. CO_2_ molecules can easily enter the network formed by associates of MEA-10H_2_O; however, during desorption, especially from cycles in which MEA molecule is included, steric hindrances may arise, since in this cycle, there are diameters commensurate with the dimensions of the CO_2_ molecule.

Theoretical spectra calculated for MEA associates with 10 water molecules have a number of frequencies that are absent in theoretical spectra of simpler associates (with 2–8 water molecules). Such frequencies, presented in Table 3, have a high intensity and very complex oscillations. These frequencies arise from the vibrational motion of terminal -OH or -NH_2_ groups of MEAs, together with associated neighboring water molecules.

The theoretical frequencies presented in Table 3 are comparable with the experimental bands in the IR spectra of 12% MEA solution, registered at a temperature of 90 °C. They increase in intensity when this sample is maintained at 90 °C (Figure 7 and Figure 8). Figure 7 depicts IR spectra of 12% MEA solution, registered at elevated temperatures, demonstrating that as the temperature increases, the intensity of bands in the 3200–3600 cm^−1^ region decreases, and new weak bands appear in the long-wavelength region of the spectrum. These spectra were recorded in attenuated total reflectance (ATR) mode for a drop of solution placed on a heated console table, i.e., part of water evaporated from solution with an increase in temperature, which causes an increase in concentration of MEA, accompanied by an increase in the intensity of absorption bands of MEA and its strong associates, and a decrease in the intensity of bands from water. For comparison, the same figure shows the spectrum of a 30% aqueous solution of MEA, recorded at room temperature, which also shows new bands recorded in the spectrum of a 12% solution at 90 °C. The differential spectrum shown in Figure 8, obtained by subtracting the spectrum of water from the spectrum of a 30% aqueous solution of MEA, makes it possible to more clearly detect the bands associated only with MEA and its associated complexes. These bands, together with certain bands characterizing the structure of the MEA molecule, are presented in Table 3, and are comparable with theoretical spectra of MEA-10H_2_O associates. Analyzing data on the association energies of associates of various structures and the data in Table 4, on the number of water molecules per MEA molecule in aqueous solutions of various concentrations, for which IR spectra were recorded, we conclude that in low-concentration solutions (up to 20%), the probability of the formation of energy-intensive and highly stable associates of MEA-10H_2_O is much lower than in solutions with concentrations of 20% and above.

In slightly concentrated solutions at low temperatures (up to 80 °C), weak associates with a low association energy of 4–8 molecules (MEA-water) will be realized. The tendency to form stronger associates (lines 9, 10 in Table 2) due to the energy gain will be reduced to the formation of purely water associates (Line 9 in Table 2), which is due to a deficiency of MEA molecules in solution and a high-value (59.7 kcal/mol) association energy. In MEA solutions of this composition, any acid gas (CO_2_, SO_2_, CS_2_, etc.) will be easily sorbed and desorbed. In more concentrated MEA solutions (from 20 to 30%), the probability of formation of MEA-10H_2_O associate complexes increases significantly due to the implementation in such solutions of a close molar ratio of MEA:H_2_O and, most importantly, due to the gain in the association energy of such a complex (67.7 kcal/mol). In the presence of such strong associates, the acid gas sorption capacity of solution will not decrease, but the gases inside such associates will be retained for longer, due to their more constrained tricyclic structure (Figure 5), which hinders their ability to desorb gases. In addition, as discussed above, the MEA nitrogen atom in such an associate complex has a tetrahedral environment, which is characteristic of ammonium cations. Upon contact with CO_2_, the formation of a charge–transfer complex is possible, followed by the formation of an ammonium cation and a carbamate anion.

Thus, the study of aqueous solutions of MEA with various concentrations, carried out by a combination of experimental and theoretical methods of vibrational spectroscopy and quantum chemistry, allows us to draw the following conclusions:MEA molecules with two hydrophilic groups have a high propensity to form associates with each other and with water molecules.In aqueous solutions of MEA with a concentration of 7 to 30%, a large number of associate complexes are realized through MEA-H_2_O hydrogen bonds. The stability and lifetime of these bonds depend on the composition of the associate and on the number and the length of hydrogen bonds.In aqueous solutions, where the concentration of which does not exceed 22%, primarily small associates are realized, in which MEA molecules are isolated from each other, due to their environment featuring energy-intensive associates from water molecules. In such solutions, both the absorption of carbon dioxide and its desorption can easily take place.In aqueous solutions of MEA with a concentration above 22%, the probability of formation of very energy-intensive (67.7 kcal/mol) complexes of MEA-10H_2_O associates increases, the structure of which makes it possible to retain CO_2_ more firmly, i.e., it does not interfere with the absorption of acid gases, but hinders their desorption.In IR spectra, the bands characterizing the energy-intensive associates of MEA-10H_2_O were identified (Table 3) and the conditions for the occurrence of such associates in aqueous solutions of MEA were established.

## 3. Materials and Methods 

### 3.1. Materials and Characterization

A stand for obtaining homogeneous transparent aqueous solutions of MEA with various concentrations and sequential saturation of these solutions with carbon dioxide was developed and modified. Using this stand, transparent solutions of MEA in distilled water of six different concentrations in the range of 7.5–30 wt.% were prepared. For the preparation of solutions, MEA of chemically pure grade was used; the mass fraction of the main substance was 99.6%.

### 3.2. FTIR-Spectroscopy, High-Temperature ATR-FTIR Spectroscopy

IR spectra were recorded on an IFS-66-v/s vacuum FT-IR spectrometer (Bruker) in transmission mode (liquid cells with CaF_2_ glasses), using a heated cell and on a Vertex 70 spectrometer (Bruker, Billerica, MA, USA), using a special high-temperature “Platinum ATR” attachment (crystal diamond). The spectra were processed using the OPUS-7 software package.

### 3.3. Quantum-Chemical Calculations

Quantum-chemical calculations were performed using the Density Functional Method (DFT B3LYP/6–31++G(d,p)) and the Hartree–Fock Method (6–31++g(d,p)) in the Gaussian software package, with full optimization of all geometric parameters of the calculated models in two media (vacuum and water), providing the calculations to obtain theoretical vibrational spectra.

## Figures and Tables

**Figure 1 molecules-28-00403-f001:**
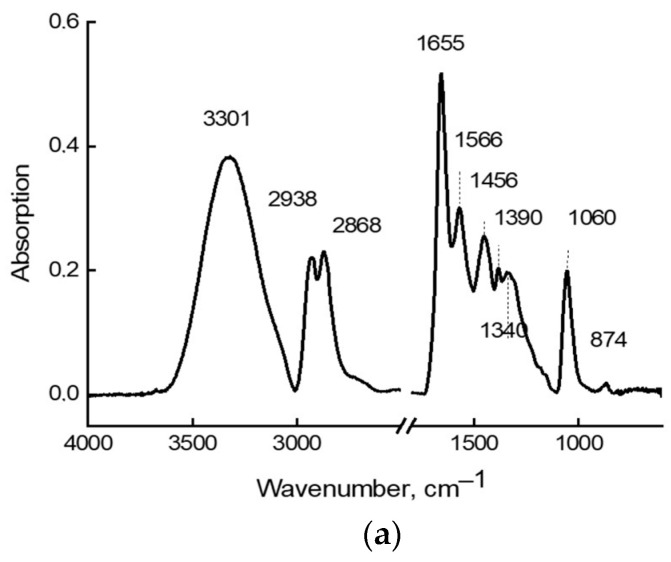
IR absorption spectra of (**a**) anhydrous MEA; (**b**) MEA aqueous solutions of various concentrations (12%, 15%, 20%, and 30%, respectively) recorded at room temperature.

**Figure 2 molecules-28-00403-f002:**
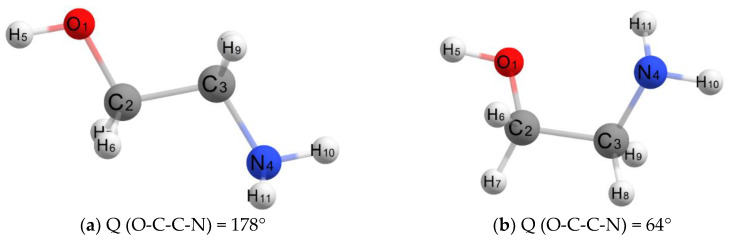
Models of two isomers of individual MEA molecules (transoidal (**a**) and gauche (**b**) structures).

**Figure 3 molecules-28-00403-f003:**
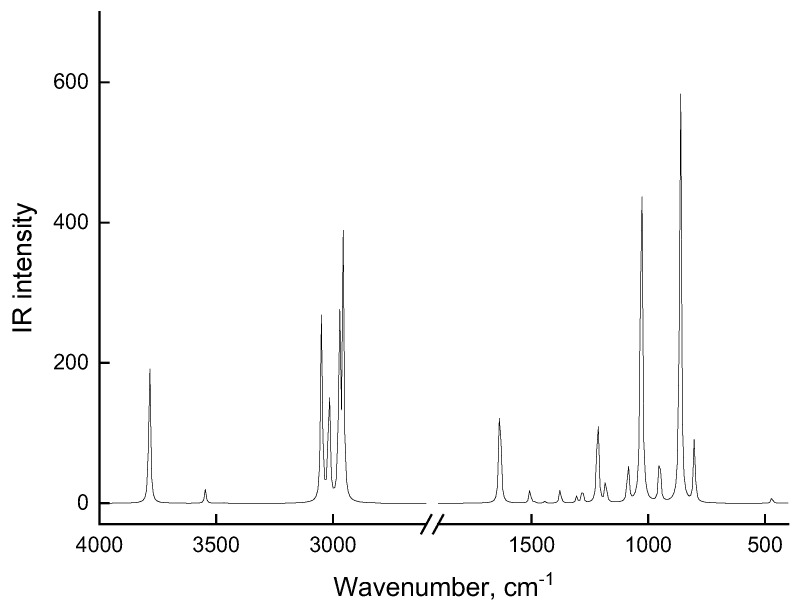
Theoretical IR spectrum of a flat transoidal model of the MEA.

**Figure 4 molecules-28-00403-f004:**
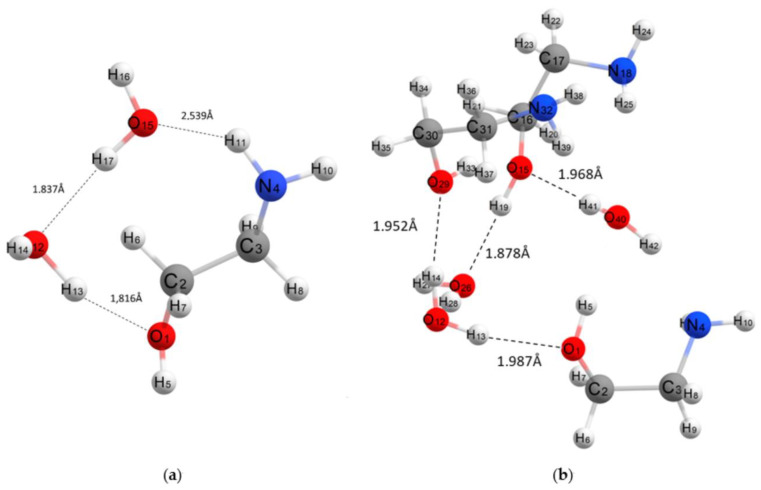
Structure of associate complexes: (**a**) MEA-2H_2_O, (**b**) 3 MEA-3H_2_O.

**Figure 5 molecules-28-00403-f005:**
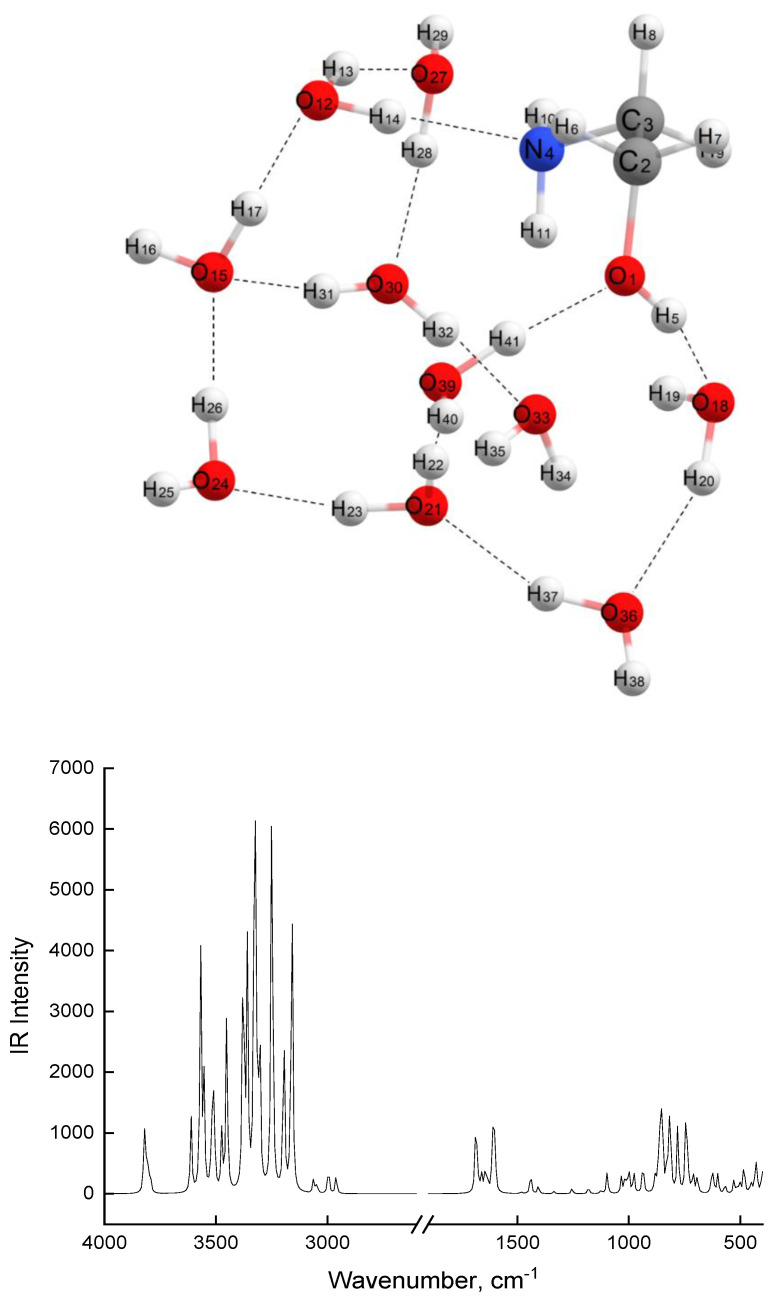
Model of MEA-10 H_2_O complex and theoretical IR spectrum of MEA-10H_2_O model.

**Figure 6 molecules-28-00403-f006:**
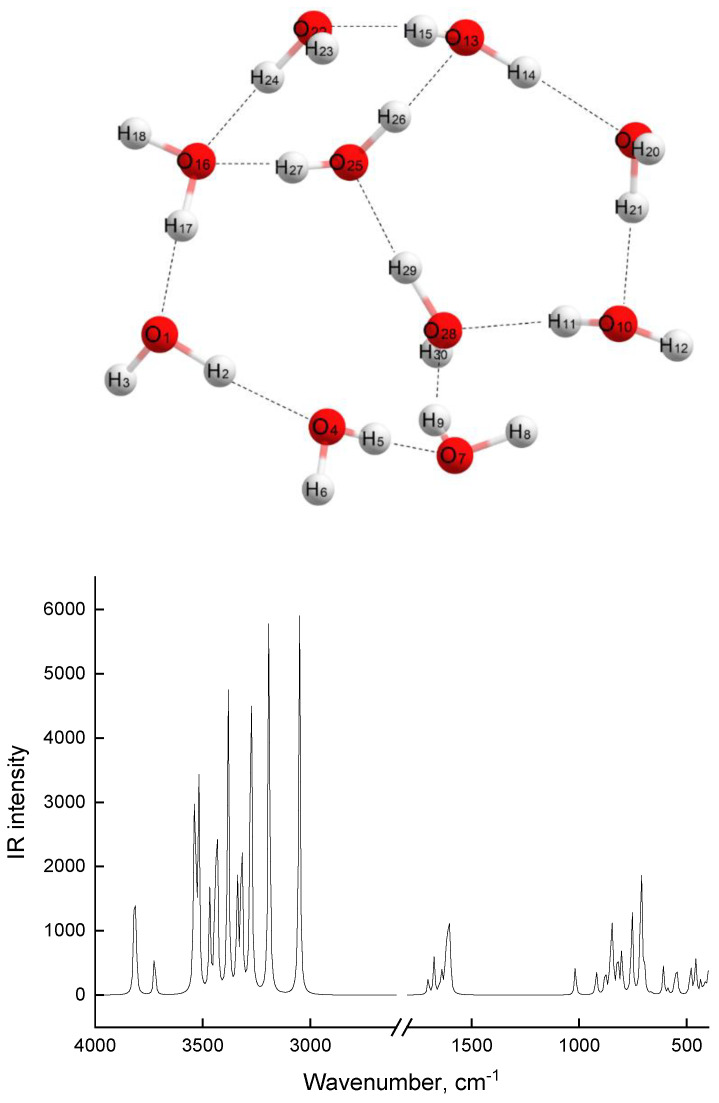
Model of 10H_2_O associate complex and theoretical IR spectrum of 10H_2_O model.

**Figure 7 molecules-28-00403-f007:**
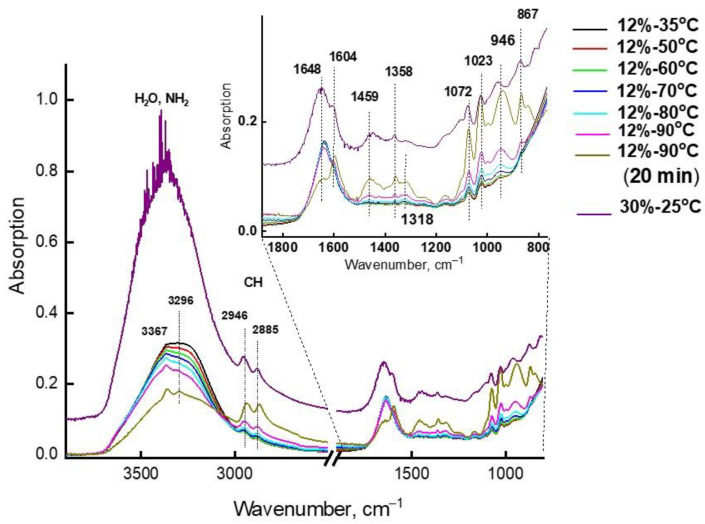
High-temperature IR absorption spectra of 12% aqueous MEA solutions: 1–35 °C, 2–50 °C, 3–60 °C, 4–70 °C, 5–80 °C, 6–90 °C, 7–90 °C after 20 min; 8-MEA (30 %) at 25 °C.

**Figure 8 molecules-28-00403-f008:**
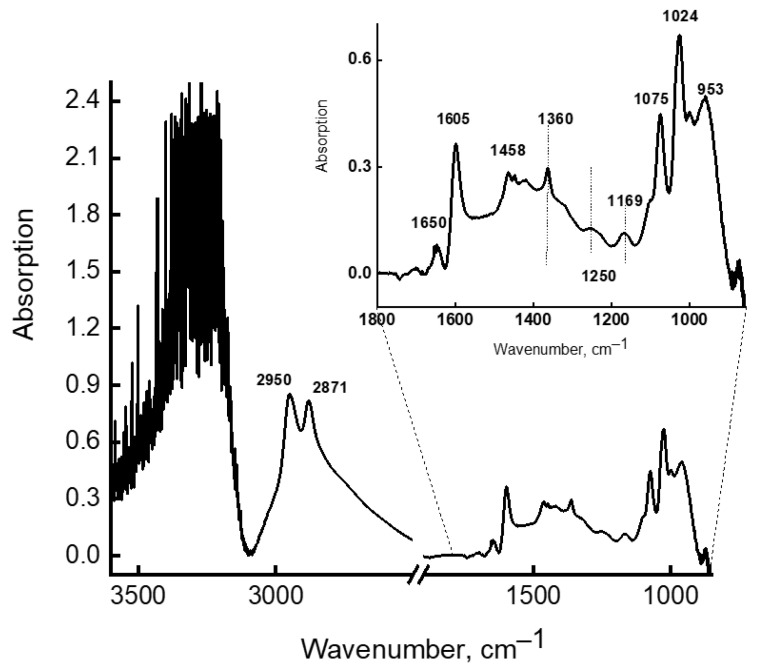
Differential spectrum of MEA (30%) registered at 90 °C.

**Table 1 molecules-28-00403-t001:** Hydrogen bond lengths and association energies in complexes Figure 4.

Associate Model	E_∆,_ kcal/mol	Hydrogen Bond Lengths (Å)	
**MEA** ** ⋯ ** **2H_2_O**	−9.11	MEA⋯H_2_O	1.816 (O_1_-H_13_)
		H_2_O⋯H_2_O	2.539 (O_15_-H_11_)
**3MEA⋯3H_2_O**	−17.87	MEA⋯H_2_O	2007 (N4-H42)
			1.987 (O1-H13)
			1.952 (O29-H14)
			1.968 (O15-H41)
			1.878 (O26-H19)
		H_2_O⋯H_2_O	1.904 (O12-H28)
			3.092 (O26-H14)
		MEA⋯MEA	2.022 (O_15_-H_33_)

**Table 2 molecules-28-00403-t002:** Structural, electronic, and energy characteristics of models of various associates in comparison with the initial models of MEA and H_2_O. An analysis of data presented in Table 2 allows us to conclude that association energy, which determines the stability of a complex, depends significantly on the structure of the complex and, to a large extent, on the number of hydrogen bonds. As the strength of the associate increases, the polarizability of N-H and O-H atoms involved in association increases, i.e., positive charges on hydrogen atoms and negative charges on oxygen and nitrogen atoms involved in the association increase. Lengths of hydrogen bonds in the most stable complexes are shorter, while bonds between atoms inside the molecule are lengthened. From the data in Table 2, it is obvious that all listed changes in structural and electronic characteristics are more pronounced in the MEA-10H_2_O associate, as well as in the associate of 10 water molecules (last 2 rows of Table 2). Structures of these associates, together with the theoretical vibrational spectra, are shown in Figure 5 and Figure 6. In these associates, oxygen atoms arise in water molecules with a negative charge equal to −1, and the total charge of two protons at such oxygen atoms is equal to +1.

No.	Associate Model	E_∆_. kcal/mol	Bond Lengths (Å)					Charges on Atoms (e)						Angle H-N-H_o_	Dip. Moment (D)
			MEA		H_2_O:O-H	HO⋯H	HN⋯H	MEA				H_2_O			
			O-H	N-H				N	H_N_	O	H_o_	O	H		
**1**	MEA	-	0.967	1.017	-	-	-	0.60	+0.31	−0.54	+0.37	-	-	106.3	2.232
**2**	H_2_O	-	-	-	0.967	-	-	-	-	-	-	-	+0.38	-	2.456
**3**	2 MEA	−3.73	0.950	1.00	-	-	2.090	−0.78	+0.31	−0.68	+0.36	-	-	106.9	0.516
**4**	3 MEA	−3.28	0.945	1.002	-	-	2.281	−0.78	+0.31	−0.68	+0.36	-	-	106.5	7.120
**5**	2H_2_O	−5.02	-	-	0.948	2.040	-	-	-	-	-	−0.76	+0.42	-	3.296
**6**	MEA-H_2_O	−5.51	0.943	1.000	0.949	2.001	-	−0.63	+0.29	−0.63	+0.37	−0.78	+0.44	107.6	2.884
**7**	MEA-2H_2_O	−9.11	0.968	1.020	0.983	1.815	2.359	−0.61	+0.33	−0.57	+0.40	−0.84	+0.45	106.9	3.589
**8**	3 MEA-3H_2_O	−17.87	0.955	1.002	0.957	1.878	2.007	−0.77	+0.30	−0.72	+0.38	−0.75	+0.41	106.2	3.650
**9**	10H_2_O	−59.70	-	-	0.995	1.645	-	-	-	-	-	−1.06	+0.66	-	7.660
**10**	MEA-10H_2_O	−67.74	0.988	1.020	0.992	1.693	1.890	−0.66	+0.45	−0.94	+0.49	−1.00	+0.50	**109.1**	2.825

**Table 3 molecules-28-00403-t003:** Comparison of theoretical frequencies of associate-complex MEA-10H_2_O with the experimental spectra (Figure 7 and Figure 8) and their attribution.

No.	Theoretical Frequency, cm^−1^	Intensity Con. UNITS	Infrared Band, CM^−1^	Attribution
**1 ***	27	0.8	-	δOH (H_2_O)
**2**	32	4.9	-	-“-
**3**	44	1.1	-	-“-
**4**	47	3.0	-	-“-
**5**	52	3.9	-	-“-
**1**	858 + 865	293	867 m *	H_2_O⋯H_2_O⋯H_2_O
**2**	947	166	946 m	H_2_O⋯H_2_O⋯NH_2_
**3**	1027	230	1024 m	NH_2_⋯ H_2_O + OH⋯OH_2_
**4**	1080	110	1075 m	H_2_O ⋯H_2_O⋯H_2_O⋯NH_2_
**5**	1147	14	1169 m	δNH_2_ + OH⋯OH_2_
**6**	1268	56	1250 m	δNH_2_-CH_2_
**7**	1352	12	1358 m	δCH_2_
**8**	1456	112	1458 m	δCH_2_
**9**	1618	252	1605 m	H_2_O⋯H_2_O⋯H_2_O
**10**	1653	78	1650	NH_2_⋯H_2_O

* The first five theoretical frequencies have positive values, i.e., they are not “imaginary”, which confirms the presence of the molecule (the associate complex MEA-10H_2_O) in the energy global minimum.

**Table 4 molecules-28-00403-t004:** The ratio of % concentration of MEA in water and the number of water molecules per 1 molecule of MEA in these solutions.

Concentration (%) MEA	7	12	15	20	30
Number of H_2_O molecules per 1 MEA molecule	45.0	24.8	19.2	13.6	7.9

## Data Availability

Not applicable.

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
