# Peer review of "Structural Features of Monoethanolamine Aqueous Solutions with Various Compositions: A Combined Experimental and Theoretical Study Using Vibrational Spectroscopy"

_molecules, 2023, doi:10.3390/molecules28010403_

Round 1

Reviewer 1 Report

This work presented by Legkov and co-workers conducted a combination of experimental FTIR and quantum chemical calculations on the monoethanolamine (MEA) structures in order to better understand the behavior of MEA in high temperature. The overall design of this study is acceptable, however, a few major issues need to be reconsidered before recommending this manuscript for publication.

1. About computational methods

In the computational part, the authors tried to optimize a few structures. It is well-known that the potential energy surface of cluster structures is quite complicated and this reviewer wonders whether all structures have been optimized to local minimum (i.e. no imaginary frequencies in vibrational analysis). Therefore, please also compile the 5 lowest vibrational frequencies calculated for all calculated structures into supporting information. And the Cartesian coordinates for all structures should also be put into SI so that readers may be able to reproduce your results. 

Secondly, the authors seemed to fully overlook the vibrational frequency scaling factors developed by Truhlar and co-workers in order to make calculated vibrational frequencies match the experimental values. The authors are advised to check on Truhlar's work [1] J. Chem. Theory Comput. 2010, 6, 2872 and [2] https://comp.chem.umn.edu/freqscale/version3b2.htm. 

2. About Figure 3 

Figure 3 is calculated IR spectrum of flat transoidal structure of MEA in gas phase. This result has two major issues. First, the authors totally overlooked the other MEA structure shown in Figure 2, and the final calculated IR spectrum should be an weighted-averaged spectrum from two structures. Second, Figure 1a is a result of MEA aggregates instead of MEA in gas phase. Therefore, the direct comparison of Figure 1a and Figure 3 is misleading. The correct approach should be to model MEA aggregates without water and calculate their IR spectra to compare against Figure 1a. 

In this regard, the authors are advised to check [1] Russian Journal of Physical Chemistry A 2016, 90, 809 and [2] Chemosphere 2020, 243, 125323. 

3. Concerning exacting the vibration information about MEA structure in their water aggregates, the authors are encouraged to try the GSVA method (Theor Chem Acc. 2021, 140, 31). 

4. This manuscript lacks the 'Conclusions' section and the readers cannot see the final conclusion of this work whether the calculated results are reasonable or not. 

5. Minor issues

line 12 and 30, should 'flue' be corrected into 'fuel'?

line 42 and 44, references [13] and [14] might need some words in advance

line 58 and 60, what does 'stand' mean? standard or protocol? 

Author Response

We thank the reviewer for their careful reading of our article, their comments and advice, which we will certainly take into account in future work. In the attachment you will find answers to your comments.

Reviewer 2 Report

The present paper deals with comparative experimental and theoretical study of vibrational spectra of solutions of small molecule monoethanolamine. The experimental IR spectra of MAE solutions have been compared with the theoretically calculated vibrational spectra of MAE-water assemblies. The calculations resulted in assignation of bands and observations on behavior of MEA associates in water solutions.

Following questions should be addressed prior to publication:

1.       Although the approach seems to be straightforward, the details on the theoretical framework are scarce. Which calculations have been performed by DFT and which by HF method? Besides the shown water molecules, was there any implicit solvent model included? How the association energies were calculated?

2.       Could you briefly discuss the observed differences between experimental and theoretical spectra of MAE solutions? Also, have you tried some different basis set, did you get better or worse agreement with experiment?

3.       In Figure 4a MAE is trans, and in 4b it is cis, isnt it? Is it a result of full geometry optimization or it was set in the input geometry?

4.       Please provide separate Discussion section after Results, it is a bit difficult to track long Results and discussion.

5.       Are there some similar theoretical studies dealing with behavior of MAE of other small molecules in solutions? If yes, please cite and compare to your theoretical model.

Author Response

(The authors gave the same response as above.)

Reviewer 3 Report

The manuscript titled: "Structural features of monoethanolamine aqueous solutions with various compositions: a combined experimental and theoretical study using vibrational  spectroscopy" by Legkov et al. reports on aqueous solutions of monoethanolamine with various concentrations at elevated temperatures. For that the method based on a comparison of experimental IR spectra and theoretical vibrational spectra of corresponding models obtained from quantum-chemical calculations have been developed.
The presented research results are valuable research material, and the work is interesting and presented in a transparent manner. Therefpre, the manuscript is suggested for the publication.

Author Response

We thank the reviewer for their careful reading and appreciation of our work.

Round 2

Reviewer 1 Report

no further reviewing is needed

Reviewer 2 Report

In the revised version the authors have properly addressed all my questions. I consider the manuscript is now suitable for publication.